# Disruption of T-box transcription factor *eomesa* results in abnormal development of median fins in Oujiang color common carp *Cyprinus carpio*

**Shiying Song**[1,2☯], **Bobo Du**[1☯], **Yu-Wen Chung-Davidson**[3], **Wenyao Cui**[1], **Yaru Li**[1], **Honglin Chen**[1], **Rong Huang**[4], **Weiming Li**[3], **Fei Li**[5]*, **Chenghui Wang**[1]*, **Jianfeng Ren**[1]*

**1** Key Laboratory of Freshwater Aquatic Genetic Resources Certificated by the Ministry of Agriculture and Rural Affairs, Shanghai Ocean University, Shanghai, China, **2** Guizhou Key Laboratory of Agricultural Biotechnology, Guizhou Institute of Biotechnology, Guizhou Academy of Agricultural Sciences, Guiyang, China, **3** Department of Fisheries and Wildlife, Michigan State University, East Lansing, MI, United States of America, **4** Guangdong Aquarium Association, Guangzhou, China, **5** Key Laboratory of Freshwater Aquaculture Genetic and Breeding of Zhejiang Province, Zhejiang Institute of Freshwater Fisheries, Huzhou, China

☯ These authors contributed equally to this work.
* jfren@shou.edu.cn (JR); wangch@shou.edu.cn (CW); lifeibest1022@163.com (FL)

## Abstract

Median fins are thought to be ancestors of paired fins which in turn give rise to limbs in tetrapods. However, the developmental mechanisms of median fins remain largely unknown. Nonsense mutation of the T-box transcription factor *eomesa* in zebrafish results in a phenotype without dorsal fin. Compared to zebrafish, the common carp undergo an additional round of whole genome duplication, acquiring an extra copy of protein-coding genes. To verify the function of *eomesa* genes in common carp, we established a biallelic gene editing technology in this tetraploidy fish through simultaneous disruption of two homologous genes, *eomesa1* and *eomesa2*. We targeted four sites located upstream or within the sequences encoding the T-box domain. Sanger sequencing data indicated the average knockout efficiency was around 40% at T1-T3 sites and 10% at T4 site in embryos at 24 hours post fertilization. The individual editing efficiency was high to about 80% at T1-T3 sites and low to 13.3% at T4 site in larvae at 7 days post fertilization. Among 145 mosaic $F_0$ examined at four months old, three individuals (Mutant 1–3) showed varying degrees of maldevelopment in the dorsal fin and loss of anal fin. Genotyping showed the genomes of all three mutants were disrupted at T3 sites. The null mutation rates on the *eomesa1* and *eomesa2* loci were 0% and 60% in Mutant 1, 66.7% and 100% in Mutant 2, and 90% and 77.8% in Mutant 3, respectively. In conclusion, we demonstrated a role of *eomesa* in the formation and development of median fins in Oujiang color common carp and established an method that simultaneously disrupt two homologous genes with one gRNA, which would be useful in genome editing in other polyploidy fishes.

**Data Availability Statement:** All relevant data are within the paper and its Supporting Information files.

**Funding:** NO. The funders had no role in study design, data collection and analysis, decision to publish, or preparation of the manuscript. This work was financially supported by the Open Project Foundation from Key Laboratory of Freshwater Aquaculture Genetic and Breeding of Zhejiang Province, Zhejiang Institute of Freshwater Fisheries (ZJK202112); Development of Genetic Improvement Technique for Ornamental Fish (D-8006-19-0166), and the SHOU&MSU Joint Research Center grant.

**Competing interests:** The authors have declared that no competing interests exist.

## Introduction

Fishes rely on fins for locomotion and balance. Their paired fins (pectoral and pelvic fins) are derived from median fins (dorsal, anal, and caudal fins). Through morphological modifications, paired fins give rise to appendages such as limbs or wings during vertebrate evolution [1]. In the vertebrate lineage, paired fins, limbs, and wings are considered as homologous organs, and their developmental processes and genetic regulatory networks are largely conserved [2,3]. Examples can be found in the studies of limbs in mice (*Mus musculus*), wings in chickens (*Gallus gallus*), and paired fins in zebrafish (*Danio rerio*) [4]. Unlike paired appendages (or paired fins) of fishes and tetrapods, median fins are present only in fishes and their developmental mechanisms are largely elusive. Nonetheless, studies have shown that median and paired fins share similar developmental mechanisms despite their different embryonic origins, namely, somatic mesoderm and lateral plate mesoderm, respectively [3,5,6].

T-box family proteins are a group of transcription factors that contain DNA-binding T-box domains, and play critical roles in limb development [7]. The T-box transcription factors are classified into five subfamilies including T, Tbx1, Tbx2, Tbx6 and Tbr1 [8], of which four subfamilies are expressed in developing limbs [7]. Within the Tbx2 subfamily, Tbx5 and Tbx4 are conserved markers for limb bud initiation and formation of the forelimb and hindlimb in mouse and chicken, respectively [9–11]. In zebrafish, deletion of *tbx5* and *tbx4* genes ceases the formation of pectoral fins and pelvic fins, respectively [10,12].

A well examined member of the Tbr1 family is Eomesodermin (Eomes), or Tbr2. In mouse limbs, *eomes* is exclusively expressed in the mesenchyme at the base of digit 4, and loss of its expression is associated with the loss of digit 4 [13]. *Eomes* is required for trophoblast development and mesoderm formation, and the embryonic development of mouse *eomes* mutant is arrested soon after implantation several days before limb formation [14]. Therefore, the exact role of *eomes* playing in limb development remains to be clarified [7]. The zebrafish genome contains two homologous *eomes* gene, *eomesa* and *eomesb* [15]. A nonsense mutation of *eomesa* in zebrafish (*eomesa^{fh105}*) generates a truncated protein in the T-box domain and results in the delay of early embryonic development and loss of the dorsal fin [16]. *Eomesb* in zebrafish is first discovered in the lymphatic system, and its protein function has not been reported yet [15].

In addition to the protein-coding gene *eomes*, several regulatory elements also affect median fin development. A long-range limb-specific enhancer, the zone of polarizing activity (ZPA) regulatory sequence, termed ZRS, controls *sonic hedgehog* (*shh*) gene expression that is critical for normal limb development [17]. A single-nucleotide mutation within ZRS causes limb malformations in multiple vertebrate species including humans [18]. Sequential nucleotide substitutions or deletions within ZRS region contribute to functional degeneration *in vivo* and result in phenotypic variations in snake body plan from a basal form with vestigial limbs to more advanced forms without limbs [19]. Deletions of both ZRS and shadow ZRS in Japanese medaka (*Oryzias latipes*) abolish *shh* expression and eliminate pectoral fin formation, whereas deletions of ZRS result in complete ablation of the dorsal fin [20]. These findings indicate that a ZRS-*Shh* regulatory module is shared by paired and median fins and that paired fins likely emerged by duplication and co-option of the developmental programs established in the median fins, supporting the notion that paired fins originated from the median fins with shared developmental mechanisms [20].

The CRISPR/Cas9 system has been widely applied to induce mutagenesis in a variety of model organisms, such as the yeast (*Saccharomyces cerevisiae*) [21], *Caenorhabditis elegans* [22], *Drosophila* [23], zebrafish [24], mouse [25] and rice (*Oraza sativa*) [26]. Furthermore, it has been shown to edit both alleles of a target gene efficiently in non-model fishes; for example,

Northeast Chinese lamprey (*Lethenteron morii*) [27], Nile tilapia (*Oreochromis niloticus*) [28], common carp (*Cyprinus carpio*) [29,30], Atlantic salmon (*Salmo salar*) [31,32], and channel catfish (*Ictalarus punctatus*) [33].

In this study, we inferred the roles of *eomesa* genes in median fin development by establishing a biallelic gene editing technology that disrupted two *eomesa* genes, *eomesa1* and *eomesa2*, in the Oujiang color common carp (*C. carpio* var. *color*). The common carp underwent a third round of whole genome duplication (4R-WGD) about 8–14.4 million years ago [34], acquiring an extra copy of protein-coding genes. The homozygotic mutants of *eomesa* in zebrafish show a phenotype of no dorsal fins. Our experiments showed that three mosaic $F_0$ founders of *eomesa1* and *eomesa2* presented the phenotype of aberrant dorsal fins and no anal fins. Therefore, the *eomesa* genes are likely involved in development of both dorsal and anal fins in Oujiang color common carp. The method to simultaneously disrupt two highly similar homologous genes provides a tool to study recently duplicated genes in other polyploidy fishes.

## Materials and methods

### Source and maintenance of color common carp

The color common carps were sampled from the Provincial Farm of Oujiang Color Common Carp at Longquan county, Zhejiang province. Sexually mature carps were raised and maintained at the Aquatic Animal Germplasm Station of Shanghai Ocean University, located in Xinchang town, Pudong New District, Shanghai. The animal sampling and experimental protocols were approved by the Shanghai Ocean University Ethics Committee for the Use of Animal Subjects.

### Identification and verification of *eomesa* genes in color common carp

Two copies of *eomesa* genes were identified in the genome of common carp. The *eomesa1* and *eomesa2* cDNA sequences of goldfish (*Carassius auratus*), Japanese silver crucian carp (*Carassius auratus* langsdorfii) and common carp downloaded from GenBank were used to identify and confirm the *eomesa* gene sequences and structures in the common carp genome (http://www.fishbrowser.org/database/Commoncarp_genome) using BLAST program (S1 Table).

The genomic sequences of common carp *eomesa1* and *eomesa2* were compared. Regions with low similarity and uniqueness to each gene were selected to design specific primer pairs to amplify *eomesa1* and *eomesa2* separately (Table 1, S1 Fig). The amplified fragments were then sequenced with the Sanger method. The sequencing results showed that these primers could be used to specifically amplify *eomesa1* and *eomesa2* of color common carp.

### Phylogenetic analysis of *eomes* genes

Deduced amino acid sequences of *eomes* genes were downloaded from the Ensembl genome browser (release 95, January 2019) for representative vertebrates, including human (*Homo sapiens*), mouse, chicken, and zebrafish. Sequence data of three other fishes were downloaded from species-specific genome databases, including grass carp (*Ctenopharyngodon idella*) (http://www.ncgr.ac.cn/grasscarp/), goldfish (https://research.nhgri.nih.gov/goldfish/), and common carp (http://www.fishbrowser.org/database/Commoncarp_genome/) (S2 Table).

Protein sequences of *eomes* gene from aforementioned species were aligned by ClustalW2 with default parameters [35]. A phylogenetic tree was built, using a JTT substitution model with maximum likelihood (ML) method with 1,000 bootstrap replications in MEGA6 software package [36].

### *Eomesa* target sites and sgRNA synthesis

Four sgRNA target sites (Table 1) were originally designed to disrupt *eomesa1* and *eomesa2* simultaneously with the ZiFiT Targeter Version 4.2 online software package (http://zifit.partners.org/ZiFiT). To avoiding sgRNA off-target effects, all candidate target sequences were aligned onto the common carp reference genome with NCBI BLAST programme and screened with the criteria of core base (12 bases adjacent to PAM) mismatches = 0. Later, the potential off-targets of the four sgRNAs were also predicted by using an CRISPR/Cas9 target online predictor CCTop with the default parameters of core length = 12, max. core mismatches = 2, max. total mismatches = 4 [37]. The results including efficacy score of each sgRNA and its targets and off-targets were listed in S2 Fig. The sgRNA synthesis was performed as described in the previous study [27] with minor modification. Briefly, the backbone from pUC19-scaffold plasmid for each gRNA was amplified by PCR with PrimeSTAR® Max DNA Polymerase (Takara) using the corresponding primers [specific forward primer: 5'-TAA TAC GAC TCA CTA TAN NNN NNN NNN NNN NNN NNN NGT TTT AGA GCT AGA AAT AGC-3', (underlined Ns represents target sequence); universal reverse primer: 5'-AAA AAA AGC ACC GAC TCG GTG CCA C-3']. The pUC19-scaffold plasmid was provided by Professor Jing-Wei Xiong of Peking University. Each target sequence begins with the GG dinucleotides, which is required for T7 RNA polymerase *in vitro* transcription. After purification of the PCR product with DNA Clean & Concentrator-5 (Zymo Research), sgRNA was synthesized using MAXIscript T7 kit (Ambion) and digested with DNase I (Ambion) for 15 min to remove DNA. Finally, the sgRNA was purified using lithium chloride precipitation method. The quality of the sgRNAs was measured by electrophoresis and bright bands (actual size 119 bp) between 100 and 150 bp (50 bp DNA Ladder, Takara) were visible on 2% agarose gel. The concentration of the sgRNAs was further measured by NanoDrop 2000 (Thermo Scientific) and concentration of the sgRNAs was about 900 ng/μl. The high quality sgRNAs was stored at -80˚C before utilized for microinjections.

## Embryo collection and microinjection

Embryos were obtained through artificial fertilization according to our previous method [29]. The eggs and sperms of color common carp were collected separately and stored at 4˚C prior to artificial fertilization.

**Table 1. The information of target sites and primer pairs for amplifying specific target sites.**

| Target ID | Target sequence | PAM sequence | Position | Gene name | Primer pairs |
|---|---|---|---|---|---|
| T1 | GGACGCGCGGAAAAGTTCTC | CGG | exon1 | *eomesa1* | ACATGGACCGGACTGAAACCGA CTGTCCGAACTGATACCCGCTC |
| | | | | *eomesa2* | ACATGGACCGGACTGAAACCGA GTCCAAACTGATAGCCGCTTCC |
| T2 | GGGCTCCGCGGCGAGGGCGC | AGG | exon1 | *eomesa1* | GACCAATCCGTGCTCTCTCTTC CGGGTTCGTTTTTATCACCCTT |
| | | | | *eomesa2* | CACCAACCCGTGCTCTCTCTTT GAACGGATTCGGGTTCATTT |
| T3 | GGCGCATTATAACGTGTTTG | TGG | exon2 | *eomesa1* | GGCAGGTGAGAATGAGAAGCTG AGTTTGCATGTAGCCTGTGTT |
| | | | | *eomesa2* | GGGCAGGTAAGAATGAAAACTT AACTGGTTGTCATGATGCTC |
| T4 | GGCTCGGTTCTTCCGCCCGC | CGG | exon1 | *eomesa1* | GGAATAAAGCAGCGGCATCCGT CGGGTTCGTTTTTATCACCCTT |
| | | GGG | | *eomesa2* | TATCCTCGACCCAACCGAGTTC GAACGGATTCGGGTTCATTT |

The embryo microinjection was conducted as our previous method with minor modification. Briefly, several glass Petri dishes were placed at the bottom of a bucket filled with water. Fertilized eggs (one-cell stage embryos) were slowly poured into the bucket. Most fertilized eggs sunk and adhered to the surface of Petri dishes with the animal pole upward, enabling precise and efficient microinjections of the mixture of sgRNA and Cas9 protein into embryos.

Each embryo was injected with about 1 nL mixture of the NLS-Cas9-NLS protein (GenScript, 800 ng/µl) and sgRNA (80–100 ng/µl). Over 500 embryos were injected with the mixture of each sgRNA and Cas9 protein. Two artificial fertilizations and embryo injection experiments were carried out on May 3 and May 11, 2018.

## Evaluation of gene editing efficiency

The exact alteration and efficiency of knockout at *eomesa1* and *eomesa2* loci was evaluated by Sanger sequencing method at two developmental time points, 24 hours post fertilization (hpf) and 7 days post fertilization (dpf). The procedure of gene editing efficiency measurement on 24 hpf embryo was diagramed in Fig 2A. For each target site, the genomic DNA from a mixture of five embryos was extracted using alkaline lysis method and then amplified with *eomesa1* and *eomesa2* locus-specific primer pairs, respectively (Table 1, S1 Fig). Fifteen colonies derived from each PCR product were sequenced and three sampling replicates were performed from embryos of each target site injection.

The procedure of gene editing efficiency measurement on 7 dpf larvae was diagramed in Fig 3A. For each target site, genomic DNA from ten larvae was extracted with TIANamp Marine Animals DNA kit (Tiangen) and then amplified with *eomesa1* and *eomesa2* locus-specific primer pairs, respectively (Table 1). PCR product from each individual was directly sequenced to determine if the target site was knocked out (Fig 3B). The knockout efficiency of ten individuals was defined as the population editing efficiency ($E_p$ = the number of knocked out individuals/total individuals examined × 100). Then, three successfully knockout individuals were randomly selected and further analyzed with PCR using TA cloning method as performed at 24 hpf. For each individual, ten independent colonies were sequenced and analyzed. The knockout efficiency of each individual was calculated and defined as the individual editing efficiency ($E_i$ = the number of knockout colonies/total colonies examined × 100). The efficiency of 7 dpf larvae was only performed on the individuals injected on May 11, 2018.

## The phenotype and genotype of mosaic $F_0$

The phenotype of $F_0$ at four months old was examined, and the fins of 145 mosaic $F_0$ fish were inspected carefully. Three individuals with dysplasia of median fins were further analyzed on their target sites with editing efficiency. The caudal fins of these individuals were used for genomic DNA extraction using a TIANamp Marine Animals DNA kit (Tiangen). DNA fragments containing four target sites were amplified using *eomesa1* and *eomesa2* locus-specific primer pairs (the primer pairs of T1 forward primer and T3 reverse primer), respectively (Table 1, S1 Fig). The target sites and their editing efficiency on each individual were then examined following the same procedure as 7 dpf (Fig 3A). To compare the phenotype presented in mosaic $F_0$ common carp with that in *eomesa* mutant zebrafish, the *eomesa*^fh105^ mutant zebrafish were obtained from China Zebrafish Resource Center and the median fin phenotypes of several generation offspring were observed and recorded.

## Results

### Characteristics of eomesa genes

Tetrapod genomes, represented here by those of chicken, mouse and human, encode an *eomes* gene while teleost genomes typically contain two homologs of the *eomes* gene, *eomesa* and *eomesb*, which are generated by the teleost-specific whole genome duplication (TS-WGD or 3R-WGD) (Fig 1A). Two paralogs of the *eomesa* gene, *eomesa1* and *eomesa2*, were encoded in the common carp genome, as a result of the fourth round whole genome duplication (4R-WGD) that occurred 14.4 million years ago (Mya), before the speciation of the common ancestor of common carp and goldfish [34,38] (Fig 1A).

The gene structures of *eomesa1* and *eomesa2* were analyzed by aligning the *eomesa* cDNA sequences of goldfish, crucian carp, and common carp onto the common carp genome assembly [34]. The *eomesa1* of common carp was located on chromosome 37, with a total length of about 7.2 kb containing six exons (S1 Table). Its CDS was 1,989 bp in length and encoded a protein of 662 amino acids which was the same length as the homolog proteins encoded in goldfish and crucian carp (S1 Table). The T-box domain of Eomesa1 was 196 amino acids in length and located at amino acid residues from 213 to 408 (Fig 1B).

The *eomesa2* of common carp was also composed of six exons and spanned about 5.1 kb on an unplaced scaffold (S1 Table). Its 1,992 bp CDS was three bp longer than that of *eomesa1* and encoded a protein of 663 amino acids. The Eomesa2 of goldfish or crucian carp was one amino acid longer than that of common carp (S1 Table). The T-box domain of Eomesa2 was located at amino acid residues from 214 to 409 and had the same length of 196 amino acids as Eomesa1 (Fig 1B). The two *eomesa* genes of common carp shared sequence identity of 94%, and their encoded proteins shared 96% identity in their amino acid residues. The alignment of Eomesa sequences from seven species showed that the amino acid sequences of T-box domain were highly conserved in vertebrates (Fig 1B).

### The efficiency of gene editing on embryos at 24 hpf

To evaluate the roles of *eomesa* in median fin development, we disrupted *eomesa1* and *eomesa2* alleles to eliminate possible functional redundancy between the two paralogous genes. Four shared target sites were designed to knock out *eomesa1* and *eomesa2* genes simultaneously (S1 Fig). Three target sites were located on exon 1 and upstream the sequences encoded the T-box domain. The fourth target site was located on exon 2 and embedded in the sequences encoding the T-box domain (Fig 1C).

The exact alteration and efficiency of knockout at the *eomesa1* and *eomesa2* loci were evaluated on 24 hpf embryos using Sanger sequencing method (Fig 2A). The knockout efficiency examined on 24 hpf embryos varied across four different target sites and between two independent experiments. In the first injection experiment, average efficiency of four targets on *eomesa1* was 29.9±20.1%, 28.9±16.8%, 19.5±9.8% and 2.2±3.9%, respectively, and on *eomesa2* was 48.9±31.5%, 34.8±26.4%, 34.8±26.4% and 7.4±7.7%, respectively (Tables 2 and S3). In the second injection experiment, the average efficiency of four targets on *eomesa1* was 51.1 ±45.4%, 40.8±24.1%, 52.8±38.8% and 12.8±6.3%, respectively, and on *eomesa2* was 40.0 ±33.3%, 48.9±43.3%, 67.5±17.3% and 16.7±5.8%, respectively (Tables 2 and S3).

The DNA sequencing results showed that many mutation types occurred, including insertion or deletion, or both (Fig 2B), which caused inframe or frame-shift mutations (S4 Table). We analyzed the mutation types and its number occurred at each target site in mosaic mutants (S4 and S5 Tables, S3 Fig). Around 20 mutation types occurred at T1 and T2 sites on *eomesa1* and *eomesa2*. More than 10 mutation types occurred at T3 site, and four and five mutation

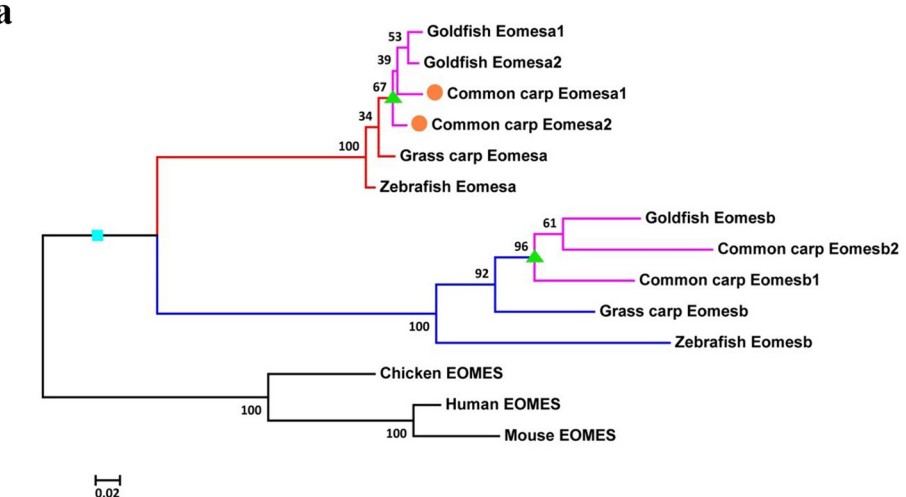

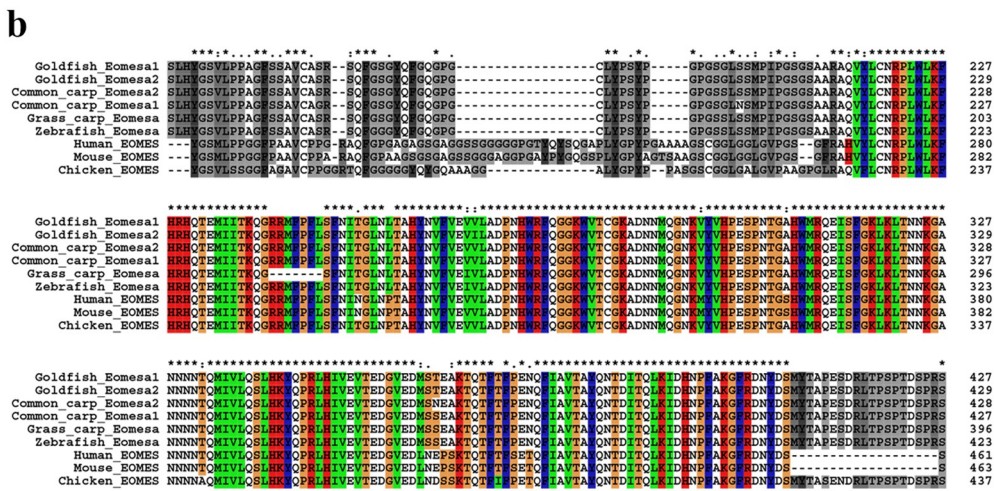

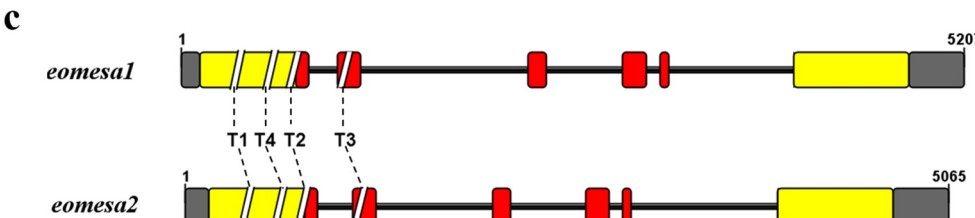

**Fig 1. The phylogenetic tree, amino acid alignment of Eomes in vertebrates, and the schematic diagrams of the *eomesa* gene structure.** (a) The phylogenetic tree was constructed using the ML method with 1,000 bootstraps. The cyan square represents teleost-specific whole genome duplication (TS-WGD, or 3R-WGD) and green triangles represent the fourth-round whole genome duplication (4R-WGD) occurred before the speciation of the common ancestor of common carp and goldfish. (b) Dashes are introduced to improve the alignment. The T-box domain is marked by colored boxes. (c) The gray boxes indicate untranslated regions. Yellow boxes indicate exons, and red boxes indicate exons encoding the T-box domain. The dash lines indicate the target sites.

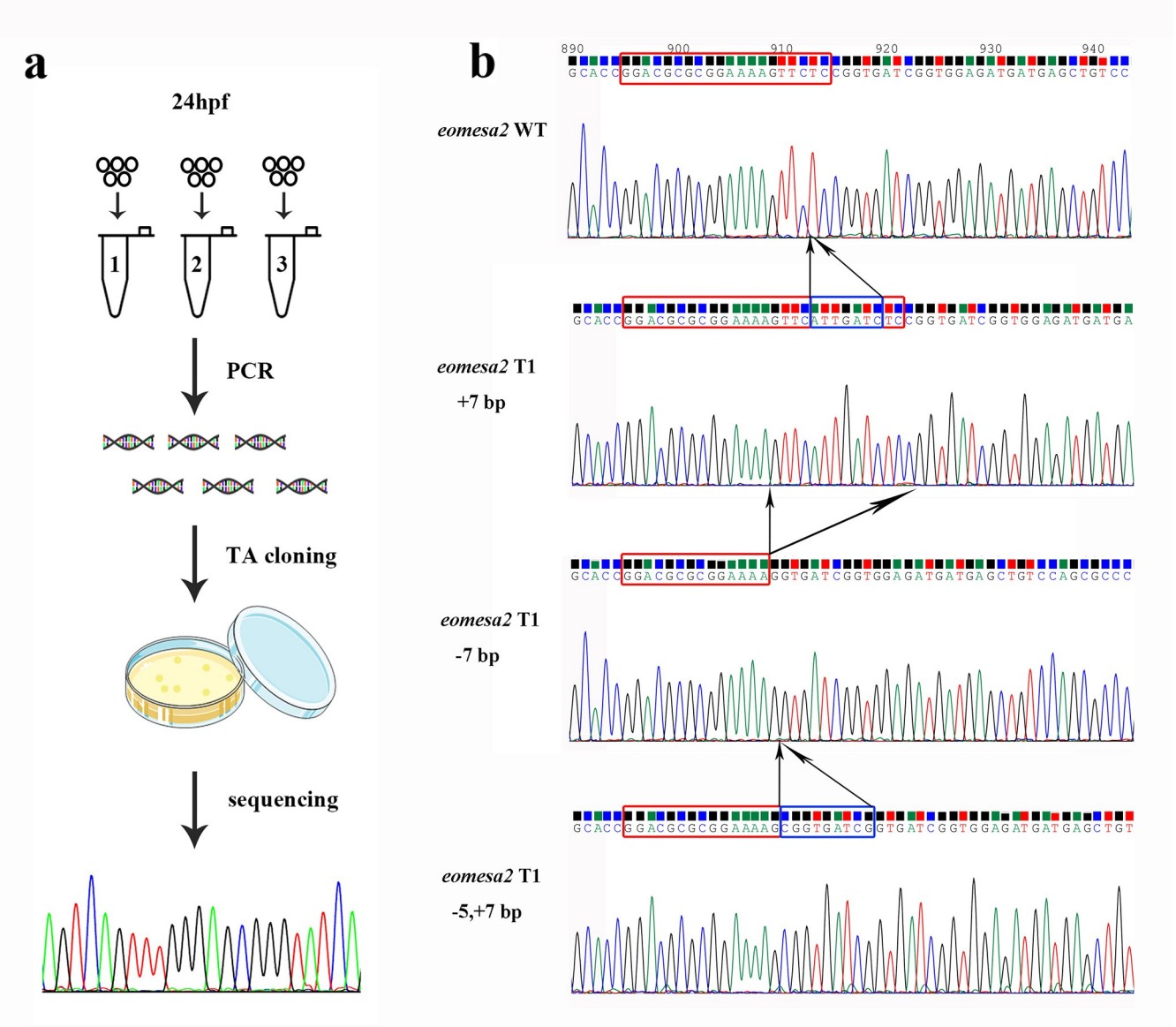

**Fig 2. The efficiency of knock out at the *eomesa1* and *eomesa2* loci was evaluated by Sanger sequencing method at 24 hours post fertilization.** (a) The procedure of gene editing efficiency measurement on 24 hpf embryo. (b) The sequencing chromatographs of TA cloning results show the representatives of three types of mutations including insertion (+7 bp), deletion (-7 bp) and both (-5,+7 bp) around T1 of *eomesa1* locus. The red rectangles indicate the target bases, and the blue rectangles indicate the inserted bases.

types occurred at T4 site, respectively. The length of DNA fragment inserted or deleted varied largely. The longest inserted fragment reached 82 nt and occurred at T4 site on *eomesa2*. The longest deleted fragment reached 65 nt and occurred at T2 sit on *eomesa1* (S4 Table).

## The efficiency of gene editing on larvae at 7 dpf

The exact alteration and efficiency of knockout at *eomesa1* and *eomesa2* in 7 dpf larvae was also evaluated using Sanger sequencing method (Fig 3A). $E_p$ on target sites from T2 to T4 was the same between *eomesa1* and *eomesa2* and was 80.0%, 70.0% and 20.0%, respectively. $E_p$ on T1 differed between *eomesa1* and *eomesa2* and was 40.0% and 30.0%, respectively (Table 3).

**Table 2. The statistics of the knockout efficiency on embryos at 24 hpf.**

| Gene | Injection times | Editing efficiency (Mean±SD) | | | |
|------|-----------------|------|------|------|------|
| | | T1 | T2 | T3 | T4 |
| *eomesa1* | 1 | 29.9±20.1% | 28.9±16.8% | 19.5±9.8% | 2.2±3.9% |
| | 2 | 51.1±45.4% | 40.8±24.1% | 52.8±38.8% | 12.8±6.3% |
| | Total | 40.5±33.5% | 34.9±19.7% | 36.1±31.2% | 7.5±7.4% |
| *eomesa2* | 1 | 48.9±31.5% | 34.8±26.4% | 34.8±26.4% | 7.4±7.7% |
| | 2 | 40.0±33.3% | 48.9±43.3% | 67.5±17.3% | 16.7±5.8% |
| | Total | 44.4±29.4% | 41.8±33.0% | 41.5±33.3% | 12.0±7.9% |

$E_i$ was evaluated by examining the sequences of ten TA colonies on each individual (Fig 3A). The average $E_i$ was calculated on three individuals (S6 Table). The average Ei on target sites from T1 to T4 at *eomesa1* was 83.3±15.3%, 76.7±15.3%, 85.2±25.6% and 13.3±11.5%, respectively, whereas $E_i$ at *eomesa2* was 76.7±15.3%, 77.5±13.9%, 50.0±26.5% and 31.2±31.8%, respectively (Table 3). These results demonstrated that in the $F_0$ mosaic population, 20 to 80 percent individuals were disrupted at an efficiency variation from 13.3% to 85.3% among different target sites. The mutation types were also examined in each individual (S4 and S7 Tables). Compared to the mutation types occurred in 24 hpf embryos, additional mutation types were found in 7 dpf larvae.

## Phenotype of $F_0$ mosaic juveniles

The phenotypes of four-month-old $F_0$ juveniles (n = 145) were examined. Three individuals, designated as Mutant 1—Mutant 3, were found to have lost the anal fin, and their dorsal fins were maldeveloped to different degrees (Fig 4B). We genotyped these three individuals and calculated the value of $E_i$ with the method used in larvae at 7 dpf. The results showed that the genomes were disrupted at T3 site in all three individuals. The $E_i$ in Mutant 1—Mutant 3 on *eomesa1* was 0%, 100% and 100%, respectively and the $E_i$ on *eomesa2* was 100%, 100% and 100%, respectively (S8 Table). The null mutation rate on *eomesa1* and *eomesa2* was further considered in each mutant as the in-frame mutations occurred. The null mutation rate on *eomesa1* and *eomesa2* was 0% and 60% in Mutant 1, 66.7% and 100% in Mutant 2, and 90% and 77.8% in Mutant 3, respectively (S8 Table).

## Discussion

### Disrupting two homologous genes simultaneously with one gRNA

The CRISPR/Cas9 system is a simple and efficient genome editing tool and has been applied to generate loss-of-function genes in model and non-model fishes. Zebrafish and Japanese medaka are two widely used diploid model fishes for study on gene function. The CRISPR/Cas9 system has been well established in these two fishes [24,39–43]. CRISPR/Cas9 system has also been used in non-model fishes, such as Nile tilapia [28], common carp [29,30], Atlantic salmon [31,32], rainbow trout (*Oncorhynchus mykiss*) [44], sea bream (*Sparus aurata*), and channel catfish [33].

Polyploidy is rarer in animals than in plants [45]. However, fishes are the most species-rich group in vertebrates, with more than 24,000 species. Polyploidy occurs in species such as common carp and goldfish in Cyprinidae, and Atlantic salmon and rainbow trout in Salmonidae [46]. Common carp is cultured in over 100 countries, providing high economic value to the global freshwater aquaculture production [47,48]. In addition to its value as a food source, common carp is widely cultured as an important ornamental fish such as koi carp because of

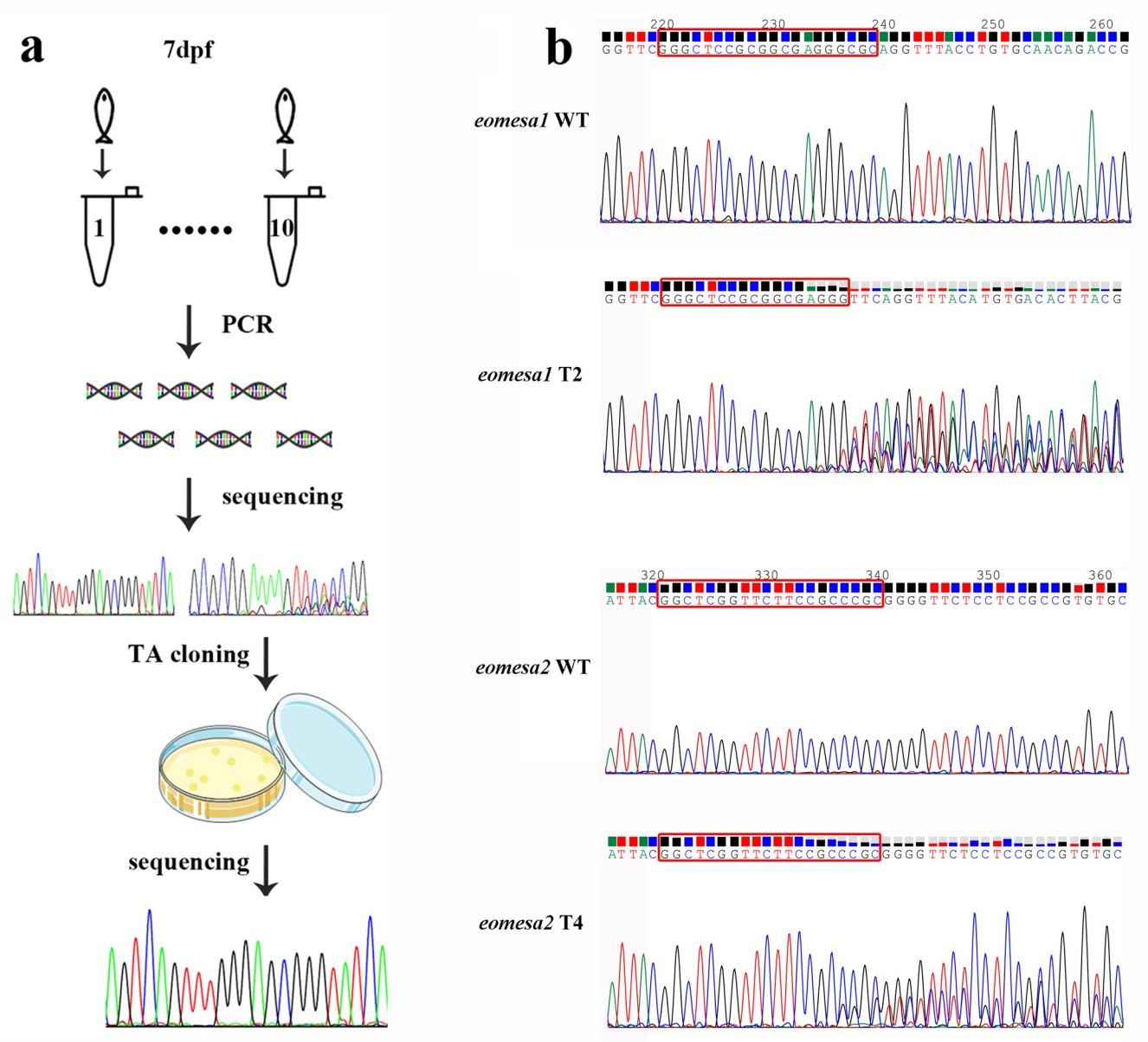

**Fig 3. The efficiency of knock out at the *eomesa1* and *eomesa2* loci was evaluated by Sanger sequencing method at 7 days post fertilization.** (a) The measurement procedure of gene editing efficiency includes population editing efficiency ($E_p$) and individual editing efficiency ($E_i$) on 7 dpf larvae. (b) Sanger sequencing chromatographs show overlapped peaks in the *eomesa1* T2 and *eomesa2* T4 sites in targeted individuals, and the ratio of peak heights from these overlapped peaks between WT and mutants implies the knockout efficiency. The red rectangles indicate the target bases.

its various color and scale patterns. However, the genetic modification of common carp still relies on artificial breeding under natural mutation, which is time-consuming and inefficient.

The CRISPR/Cas9 system provides a highly efficient targeted mutagenesis tool for common carp genetic studies and breeding. The knockout efficiency examined on 24 hpf embryos varied across four different target sites and between two independent experiments. The average knockout efficiency varied from 7.5±7.4% at T4 site to 44.4±29.4% at T1 site. The values of average knockout efficiency were similar (around 40%) at T1-T3 sites between *eomesa1* and *eomesa2*. Likewise, the average knockout efficiency was approximately 10% at T4 site for both

**Table 3. The statistics of the knockout efficiency on larvae at 7dpf.**

| Gene | Population editing efficiency ($E_p$) [†] | | | | Individual editing efficiency ($E_i$) [‡] | | | |
|---|---|---|---|---|---|---|---|---|
| | T1 | T2 | T3 | T4 | T1 | T2 | T3 | T4 |
| *eomesa1* | 40.0% | 80.0% | 70.0% | 20.0% | 83.3±15.3% | 76.7±15.3% | 85.2±25.6% | 13.3±11.5% |
| *eomesa2* | 30.0% | 80.0% | 70.0% | 20.0% | 76.7±15.3% | 77.5±13.9% | 50.0±26.5% | 31.2±31.8% |

Note: [†]The knock out efficiency on **ten** individuals was defined as the population editing efficiency ($E_p$ = the number of knock-outed individuals/total **ten** examined individuals × 100).

[‡]The knock out efficiency on each of three random individuals was calculated and its value was defined as the individual editing efficiency ($E_i$ = the number of knocked out colonies/total examined colonies × 100); here presented the average $E_i$ of three individuals.

*eomesa1* and *eomesa2* (Table 2). Similar to the results from 24 hpf, both of $E_p$ and $E_i$ at 7 dpf also varied across target sites. The $E_p$ varied from 20.0% at T4 site to 70.0–80.0% at T2 and T3 sites. The $E_i$ at T1 to T3 was as high as about 80%, while the $E_i$ at T4 site was 13.3±11.5% for *eomesa1* and 31.2±31.8% for *eomesa2* (Table 3). However, the knockout efficiency on 24 hpf embryos in experiment performed at the second time was at least 4.0% higher than that in experiment at the first time, which is probably due to improved injection skill at the second time. The introduced somatic mutation frequencies in this study are comparable to the frequencies of zebrafish [49] and common carp [30] in previous studies. The reason why the efficiency varies between different sites is unclear, but we speculate that it may be related to the local DNA three-demensional structure of the target site. A similar phenomenon has been reported in zebrafish [50].

Although the CRISPR/Cas9 technology is simple and high efficient in cleaving target regions, its off-target effects often occurred and could not be ignored. We predicted the potential off-targets of the four sgRNAs by CCTop and their efficacy scores were also calculated by CRISPRater (S2 Fig). The efficacy scores of four sgRNAs were from medium (0.63) to high (0.81). The potential off-target site positions mostly located at intergenic or intronic regions. However, the results also showed that the T2 sgRNA targeted an exonic region of *syncrip* gene and T4 sgRNA targeted an exonic region of unknown gene with core mismatches = 2 and that the T3 sgRNA targeted the exonic regions of *eomesb* genes (*eomesb1* and *eomesb2*) with core mismatches = 1 (S2 Fig). We checked whether the exonic regions of *eomesb* genes (*eomesb1* and *eomesb2*) were targeted by T3 sgRNA in the ten larvae at 7 dpf with the similar procedure used in individual gene editing efficiency measurement on 7 dpf larvae (Fig 3A). However, the sequencing results showed that exonic regions of *eomesb* genes were not targeted by T3 sgRNA (S1 File). Therefore, we speculated that the abnormal development of median fins in Oujiang common carp (Fig 4) was not caused by the loss function of *eomesb* because the gene knockout zebrafish of *eomesb* generated previously in our lab did not present any phenotypes related to fin development [51].

Generally, single gene mutant is generated through disrupting the target gene with one or more gRNAs whereas double gene mutant is generated through disrupting the target genes individually and then crossing between the offspring with individual gene mutation. Sometimes, double gene mutant is also generated through disrupting the target genes with two different gRNAs simultaneously. For example, Zhong et al. disrupted two common carp *sp7* genes, *sp7a* and *sp7b*, individually and generated double mutant fish of *sp7a;mstnba* with co-microinjection of each target gene gRNA in a single step [30]. In this study, we disrupted two homologous genes with one gRNA. To our knowledge, this is the first report on targeted disruption of two homologous genes with one gRNA in polyploidy fish. This strategy saves one generation time to obtain homozygotic double mutants compared to disrupting the target

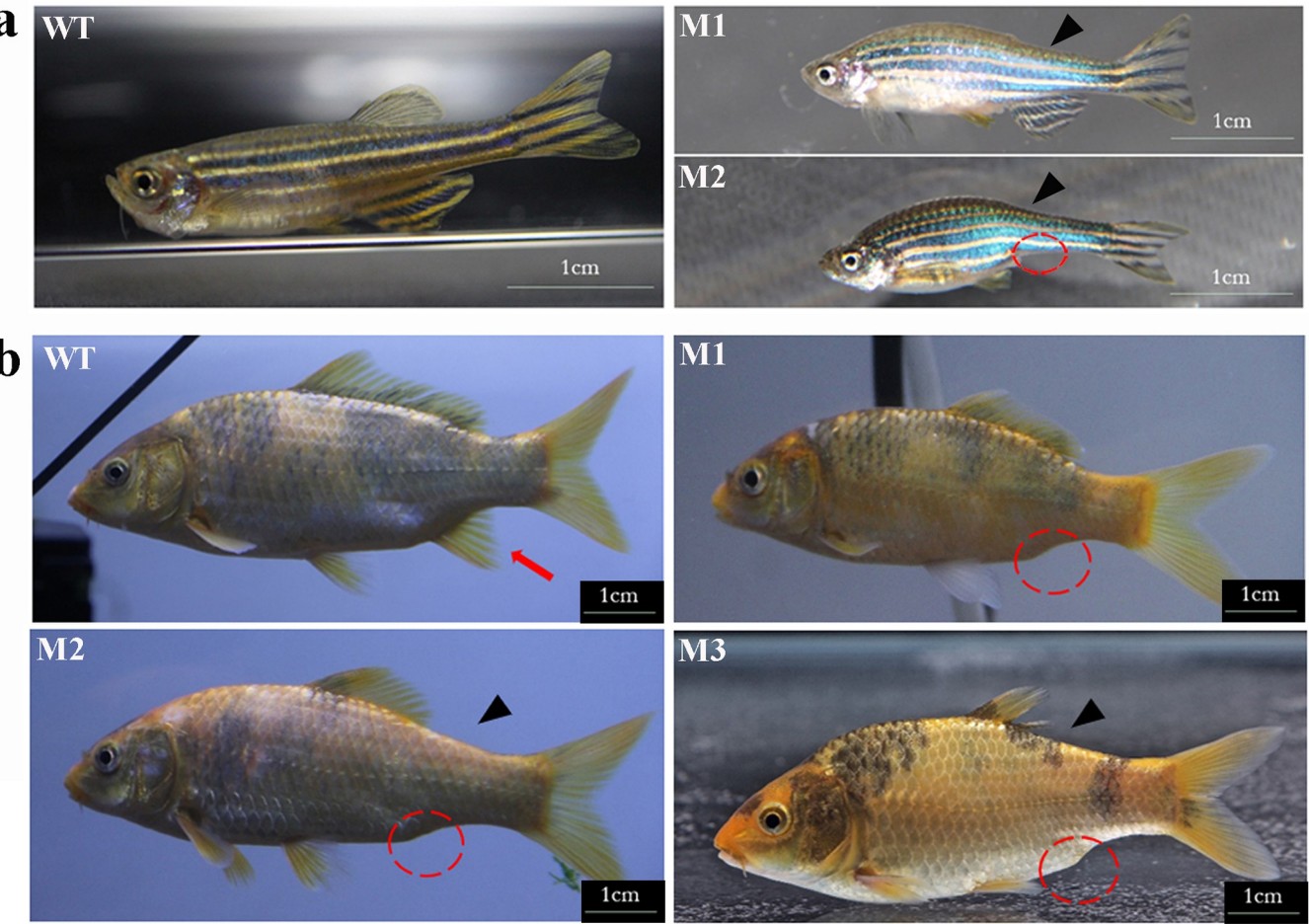

**Fig 4. Phenotypic comparison between zebrafish and color common carp *eomesa* mutants.** (a) Lateral view of wild type and *eomesa* mutant zebrafish. Compared to the wild type, the *eomesa^{fh105}* mutants exhibit two phenotypes. All mutants exhibit the phenotype with no dorsal fin. Rare mutants exhibit the phenotype with no dorsal fin and anal fin. (b) Lateral view of four-month-old wild type and *eomesa* mosaic mutants of color common carp. Compared to the wild type, three mosaic mutants (M1-M3) exhibit different degrees of deformity in the dorsal fins and complete loss of anal fins.

genes individually and then crossing between the offspring with individual gene mutation. In addition, this strategy is also beneficial to studying the evolution of duplicated genes (either neofunctionalization or subfunctionalization) in tetraploidy fish because homozygotic single gene mutants and double gene mutants will simultaneously appear in $F_2$ progeny.

## Eomesa involvement in the formation and development of median fins

When vertebrates transit from aquatic to terrestrial habitats, the median fins degenerate and the paired fins evolve into forelimbs and hindlimbs that are more adaptive to terrestrial habitats. The developmental processes and genetic regulatory networks of paired appendages have attracted much attention and have been extensively studied [4]. Although median fins play an important role in generating locomotor force during both steady swimming and maneuvering [8], their origin and developmental mechanisms have remained largely unknown. Only a few studies have shown that they share similar developmental mechanisms with paired fins [3,5,6].

*Eomesa* is the only maternal T-box transcription factor found in zebrafish that promotes mesoderm and ectoderm induction in early embryonic development and known to have an

impact on the development of dorsal fin [16]. Du et al. (2012) report that the zebrafish *eomesa^{fh105}* mutant generated by TILLING exhibits a phenotype with no dorsal fin. However, the association between *eomesa* and anal fin development has not been established in zebrafish. In our observation of median fin phenotypes of zebrafish *eomesa* mutants, we found that several mutants exhibit the phenotype with no dorsal and anal fin in one batch of *eomesa^{fh105}* progeny (Fig 4B). Our double knockout *eomesa* mutants of common carp exhibited the phenotype with dorsal fin dysplasia, anal fin loss, and normal caudal fin. This may be due to the concerted developmental and evolutionary pattern of the dorsal and anal fins. Thus, we speculate that two *eomesa* genes, *eomesa1* and *eomesa2* are functionally redundant and dosage-dependently affect the development of median fins excluding caudal fin in Oujiang common carp. The offspring with knockout of only single gene may have weak phenotype with median fin defect. We further speculate that the failure of fin bud formation in dorsal and anal fins may be caused by *eomesa* in the mesoderm cells that form fin buds in the postembryonic development. It is interesting that truncated *eomesa* proteins with different length had differential effects on the development of median fins formation. We originally planned to perform functional analysis of *eomesa* gene with the homozygotes and did not do more genotyping analysis on other mosaic fish without phenotype. Unfortunately, due to the outbreak of COVID-19 pandemic, all the mosaic $F_0$ fish raised at the farming base died in 2020. In-depth analysis on the *eomesa* gene could not be further carried out.

## Conclusion

In this work, we established a method to simultaneously disrupt two homologous genes with one gRNA and demonstrated that *eomesa* was likely involved in the formation and development of median fins in Oujiang color common carp. Our study also provides a useful and potentially efficient avenue for genome editing in polyploidy fishes.

## Supporting information

**S1 Fig. The position of four target sites and their locus-specific primer pairs.**
(DOCX)

**S2 Fig. The information of four targets and their off-target sites presented in CCTop.**
(DOCX)

**S3 Fig. The sequences of the target region of *eomesa* in the common carp larvae at 24 hpf and 7 dpf.**
(DOCX)

**S1 Table. Genomic position of *eomesa1* and *eomesa2* and comparison of *eomesa* sequences between goldfish, common carp and crucian carp.**
(XLSX)

**S2 Table. Information of species used for phylogenetic tree.**
(XLSX)

**S3 Table. The statistics of the knockout efficiency on 24 hpf embryos with three sampling replicates.**
(DOCX)

**S4 Table. Statistics of mutation types on four target sites.**
(XLSX)

**S5 Table. The statistics of the mutation types on 24 hpf embryos.**
(DOCX)

**S6 Table. The individual knockout efficiency on 7 dpf larvae with three random individuals.**
(DOCX)

**S7 Table. The statistics of the mutation types on 7 dpf larvae.**
(DOCX)

**S8 Table. Mutation types in three mosaic fish.**
(XLSX)

**S1 File. Examination of potential off-target on *eomesb* gene.**
(DOCX)

## Acknowledgments

We want to thank the editor Baisong Lu and two anonymous reviewer for their helpful and constructive comments on our manuscript.

## Author Contributions

**Conceptualization:** Jianfeng Ren.

**Data curation:** Bobo Du, Yaru Li.

**Funding acquisition:** Rong Huang, Fei Li, Jianfeng Ren.

**Investigation:** Shiying Song, Wenyao Cui.

**Methodology:** Honglin Chen.

**Project administration:** Jianfeng Ren.

**Resources:** Honglin Chen, Chenghui Wang.

**Supervision:** Jianfeng Ren.

**Writing – original draft:** Shiying Song, Bobo Du, Jianfeng Ren.

**Writing – review & editing:** Yu-Wen Chung-Davidson, Weiming Li, Chenghui Wang, Jianfeng Ren.

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
