## [Decision Letter · Decision Letter 0]

20 Jun 2022

PONE-D-22-14986Disruption of T-box transcription factor eomesa results in abnormal development of median fins in Oujiang color common carp Cyprinus carpioPLOS ONE

Dear Dr. Ren,

Thank you for submitting your manuscript to PLOS ONE. After careful consideration, we feel that it has merit but does not fully meet PLOS ONE’s publication criteria as it currently stands. Therefore, we invite you to submit a revised version of the manuscript that addresses the points raised during the review process.

In addition to the issues raised by the two reviewers, the author should take care of the following issues:1) Potential off-targets of the four sgRNAs should be predicted and listed.2) Potential targeting of homologous genes (Eomesb) by the four sgRNAs should be examined by prediction. If there is a possibility of hitting these genes, design experiments (Surveyor assay or sanger sequencing) to check editing in those sites.3) Either increase the number of sanger sequencing clones or do NGS analysis.4) In Fig.2 and Fig.3, the authors should align all mutated sequences with the reference sequence.  Indicate PAM and sgRNA target sequence. 5) In table 1, the PAM sequences should be listed.6) The authors knocked out both genes and observed phenotypes. Comparing the observed phenotypes with those after knocking out only one gene should provide information on the roles of each gene in fin development. The authors should discuss this point in the paper.==============================

We look forward to receiving your revised manuscript.

Kind regards,

Baisong Lu, Ph.D

Academic Editor

PLOS ONE

Journal Requirements:

"NO. The funders had no role in study design, data collection and analysis, decision to publish, or preparation of the manuscript."

"This work was financially supported by the Open Project Foundation from Key Laboratory of Freshwater Aquaculture Genetic and Breeding of Zhejiang Province, Zhejiang Institute of Freshwater Fisheries (ZJK202112); Development of Genetic Improvement Technique for Ornamental Fish (D-8006-19-0166), and the SHOU&MSU Joint Research Center grant."

"NO. The funders had no role in study design, data collection and analysis, decision to publish, or preparation of the manuscript."

Reviewers' comments:

Reviewer's Responses to Questions

**Comments to the Author**

1. Is the manuscript technically sound, and do the data support the conclusions?

Reviewer #1: Yes

Reviewer #2: No

2. Has the statistical analysis been performed appropriately and rigorously? 

Reviewer #1: Yes

Reviewer #2: No

3. Have the authors made all data underlying the findings in their manuscript fully available?

Reviewer #1: Yes

Reviewer #2: Yes

4. Is the manuscript presented in an intelligible fashion and written in standard English?

Reviewer #1: Yes

Reviewer #2: Yes

5. Review Comments to the Author

Reviewer #1: In this manuscript, the author trying to design a certain kind of sgRNA sequence to disrupt two homologous genes (eomesa) in polyploidy fish. The innovation of the article is acceptable, but there are some problems needs to be solved.

(1) Detection of gene editing efficiency is an important for this article, but sanger sequencing is not that convincible and the sample size of 10 could not accurately reflect its effectiveness, next generation sequence is highly recommended to verify its efficiency.

(2) The SgRNA sequence design and optimization should be incorporated into the method. And the SgRNA's quality should be verified and supplemented into the manuscript.

(3) What caused the difference of gene editing efficiency between two injections. Will this difference have an impact on the later detection of gene editing efficiency? Please explain this difference in the discussion part.

(4) In this paper, only sequencing and topography methods were used to observe the effectiveness of Gene Editing efficiency, Immunofluorescence detection (IF) or Western Blot (WB) for target protein are highly recommended to supplement in the manuscript.

Reviewer #2: In the present manuscript, Shiying Song et al aim to define the roles of eomesa genes in the median fin development in color common carp. By using gene editing and Sanger sequencing experiments, they disrupted eomesa genes and showed that eomesa genes are essential for the development of median fins in Oujiang color common carp. The biggest issue with the manuscript is that it is primarily engaged in confirming the observations made by the other studies. The manuscript has not provided further insights into the underlying mechanisms.

1. In Figure 2b, the labeling should be the eomesa1 loci.

2. In Figure 4a, it is not clear to the reviewer whether the data of the eomesa mutant in zebrafish are from the literature or generated by the authors.

6. PLOS authors have the option to publish the peer review history of their article (what does this mean?). If published, this will include your full peer review and any attached files.

Reviewer #1: No

Reviewer #2: No

---

## [Author Response · Author response to Decision Letter 0]

22 Dec 2022

Response to Reviewers’ Comments

Reviewer 1: 

In this manuscript, the author trying to design a certain kind of sgRNA sequence to disrupt two homologous genes (eomesa) in polyploidy fish. The innovation of the article is acceptable, but there are some problems needs to be solved.

Comment 1. Detection of gene editing efficiency is an important for this article, but Sanger sequencing is not that convincible and the sample size of 10 could not accurately reflect its effectiveness, next generation sequence is highly recommended to verify its efficiency.

Response: Thanks to the reviewer’s constructive suggestion. Unfortunately, due to the outbreak of COVID-19 pandemic, all the mosaic F0 fish raised at the farming base died. We checked the DNA samples stored in our lab and found only a little bit of DNA from 7 dpf larvae were remained. The quality and quantity of the remained DNA is not suit for NGS analysis. The detection of gene editing efficiency with NGS method could not be performed. We used the DNA for off-target analysis via PCR-TA clone sequencing method (see below. response to editor’s comment 2). NGS analysis is a good detection method that we have not used before. Recently, we did some gene editing work in other cultured fishes, we will try to use this method in gene editing detection. Thank you very much.

Comment 2. The SgRNA sequence design and optimization should be incorporated into the method. And the SgRNA's quality should be verified and supplemented into the manuscript.

Response: According to the reviewer’s suggestion, we inserted the sentences “To avoiding sgRNA off-target effects, all candidate target sequences were aligned onto the common carp reference genome with NCBI BLAST programme and screened with the criteria of core base (12 bases adjacent to PAM) mismatches = 0” into the section of “Materials and methods”. In the same paragraph, we described how to verify SgRNA's quality as “The quality of the sgRNAs was measured by electrophoresis and bright bands (actual size 119 bp) between 100 and 150 bp (50 bp DNA Ladder, Takara) were visible on 2% agarose gel. The concentration of the sgRNAs was further measured by NanoDrop 2000 (Thermo Scientific) and concentration of the sgRNAs was about 900 ng/μl.

Comment 3. What caused the difference of gene editing efficiency between two injections. Will this difference have an impact on the later detection of gene editing efficiency? Please explain this difference in the discussion part.

Response: In our lab, we mostly do gene editing work using the model fish zebrafish. This is the first gene editing work using the non-model fish common carp. As you known, the common carp is very different from zebrafish in the spawning frequency and it spawns once per year and its eggs have some difference with those of zebrafish in the size, viscosity, intensity of egg membrane and so on. We have no chance to practice injecting with the egg of common carp in advance. Thus, we were more skilled in the second injection experiment than in the first injection experiment. The proficiency in injection skill has an impact on the later detection of gene editing efficiency. As you see, the knockout efficiency in experiment performed at the second time was at least 4.0% higher than that in experiment at the first time. We added a sentence to explain this difference in the Discussion section as following “However, the knockout efficiency on 24 hpf embryos in experiment performed at the second time was at least 4.0% higher than that in experiment at the first time, which is probably due to improved injection skill at the second time.”

Comment 4. In this paper, only sequencing and topography methods were used to observe the effectiveness of gene editing efficiency, Immunofluorescence detection (IF) or Western Blot (WB) for target protein are highly recommended to supplement in the manuscript. 

Response: Response: The most used methods to detect gene editing efficiency in zebrafish is the combination of endonuclease digestion and gel electrophoresis, and sequencing including Sanger sequencing for low throughput screening and NextGen sequencing for high throughput screening of multiple loci. Immunofluorescence detection (IF) or Western Blot (WB) are probably good methods to detect target protein level in model animals such as mouse or rat. However, even in the model fish zebrafish, the applicable antibodies are limited. Thus, to date, the detection methods of IF and WB are still difficult to apply into common carp.

Reviewer 2: 

In the present manuscript, Shiying Song et al aim to define the roles of eomesa genes in the median fin development in color common carp. By using gene editing and Sanger sequencing experiments, they disrupted eomesa genes and showed that eomesa genes are essential for the development of median fins in Oujiang color common carp. The biggest issue with the manuscript is that it is primarily engaged in confirming the observations made by the other studies. The manuscript has not provided further insights into the underlying mechanisms.

Comment 1. In Figure 2b, the labeling should be the eomesa1 loci.

Response: We checked the Figure 2b again. The label is correct. We used the T1 site of eomesa2 locus as an example to show that many mutation types (insertion or deletion, or both) occurred.

Comment 2. In Figure 4a, it is not clear to the reviewer whether the data of the eomesa mutant in zebrafish are from the literature or generated by the authors.

Response: The phenotypic figures of wild type and eomesa mutant zebrafish were generated by our lab. We added several sentences into the section of Materials and methods as following “To compare the phenotype presented in mosaic F0 common carp with that in eomesa mutant zebrafish, the eomesafh105 mutant zebrafish were obtained from China Zebrafish Resource Center and the median fin phenotypes of several generation offsprings were observed” and one sentence into the section of Discussion as following “In our observation of median fin phenotypes of eomesa mutants”.

Response to Editor’s Comments

In addition to the issues raised by the two reviewers, the author should take care of the following issues:

Comment 1. Potential off-targets of the four sgRNAs should be predicted and listed.

Response: We predicted the potential off-targets of the four sgRNAs by using an CRISPR/Cas9 target online predictor CCTop with the default parameters of core length = 12, max. core mismatches = 2, max. total mismatches = 4. The results including efficacy score of each sgRNA and its targets and off-targets were listed in Supplemental Figure 2. The results of off-targets were further described in the section of Discussion.

Comment 2. Potential targeting of homologous genes (Eomesb) by the four sgRNAs should be examined by prediction. If there is a possibility of hitting these genes, design experiments (Surveyor assay or Sanger sequencing) to check editing in those sites.

Response: We checked the potential off-targets of the four sgRNAs predicted with CCTop. Indeed, among the four sgRNAs designed, the T3 sgRNA possibly targets the exonic regions of eomesb genes (eomesb1 and eomesb2) as there are three or four mismatches (Indicated with red rectangle in Supplemental Figure 2). We checked whether the exonic regions of eomesb genes (eomesb1 and eomesb2) were targeted by T3 sgRNA using the DNA samples of 10 larvae at 7 dpf stored at lab with PCR-TA clone sequencing method. The sequencing results showed that exonic regions of eomesb genes (eomesb1 and eomesb2) were not targeted by T3 sgRNA (The results were supplemented in Supplementary-materials-for-offtarget and discussed in the section of Discussion). Therefore, we infer that the abnormal development of median fins was not caused by the loss function of eomesb. We generated gene knockout zebrafish of emosb in previous study. It did not present any phenotypes related to fin development. We added these information and cited our published paper in Discussion.

Comment 3. Either increase the number of Sanger sequencing clones or do NGS analysis. 

Response: Thank you for your good suggestion. We checked the DNA samples stored in our lab and found only a little bit of DNA from 7 dpf larvae were remained. The quality and quantity of the remained DNA is not suit for NGS analysis. We used the DNA for off-target analysis via PCR-TA clone sequencing method. NGS analysis is a good detection method that we have not used before. Recently, we did some gene editing work in other cultured fishes, we will try to use this method in gene editing detection. Thank you very much.

Comment 4. In Fig. 2 and Fig. 3, the authors should align all mutated sequences with the reference sequence. Indicate PAM and sgRNA target sequence. 

Response: We aligned all mutated sequences with the reference sequences and indicated PAM and sgRNA target sequence. These results were listed in Supplemental Figure 3.

Comment 5. In table 1, the PAM sequences should be listed.

Response: We listed the PAM sequences for each target on eomesa1 and eomesa2.

Comment 6. The authors knocked out both genes and observed phenotypes. Comparing the observed phenotypes with those after knocking out only one gene should provide information on the roles of each gene in fin development. The authors should discuss this point in the paper.

Response: Thank your for your constructive suggestion. Unfortunately, due to the outbreak of COVID-19 pandemic, all the mosaic F0 fish raised at the farming base died. We could not obtain the offspring with knockout of only one gene and observe their phenotypes. We speculate that the functions of two eomesa genes are redundant and dosage-dependent on median fin development. The offspring with knockout of only single gene may have weak phenotype with median fin defect. We discussed this point in the paper.

---

## [Editor Report · Decision Letter 1]

20 Jan 2023

Disruption of T-box transcription factor eomesa  results in abnormal development of median fins in Oujiang color common carp Cyprinus   carpio

PONE-D-22-14986R1

Dear Dr. Ren,

We’re pleased to inform you that your manuscript has been judged scientifically suitable for publication and will be formally accepted for publication once it meets all outstanding technical requirements.

Kind regards,

Baisong Lu, Ph.D

Academic Editor

PLOS ONE
---

## [Editor Report · Acceptance letter]

7 Feb 2023

PONE-D-22-14986R1 

Disruption of T-box transcription factor *eomesa* results in abnormal development of median fins in Oujiang color common carp *Cyprinus carpio*

Dear Dr. Ren:

I'm pleased to inform you that your manuscript has been deemed suitable for publication in PLOS ONE. Congratulations! Your manuscript is now with our production department. 

Kind regards, 

on behalf of

Dr. Baisong Lu 

Academic Editor

PLOS ONE